# Sexual selection for both diversity and repetition in birdsong

Javier Sierro [1], Selvino R. de Kort [2] & Ian R. Hartley [1]

From fiddler crabs to humans, animals perform repetitive displays showing neuromotor skill and vigour. Consistent repetition of identical notes (vocal consistency) facilitates the assessment of neuromotor skills and is important in communication in birds. Most birdsong research has focused on song diversity as a signal of individual quality, which seems contradictory as repetition is extremely common in most species. Here we show that consistent repetition within songs is positively correlated with reproductive success in male blue tits (*Cyanistes caeruleus*). A playback experiment shows that females are sexually aroused by male songs with high levels of vocal consistency, which also peaks seasonally during the fertile period of the female, supporting the role of vocal consistency in mate choice. Male vocal consistency also increases with subsequent repetitions of the same song type (a warm-up effect) which conflicts with the fact that females habituate to repeated song, showing decreased arousal. Importantly, we find that switching song types elicits significant dishabituation within the playback, supporting the habituation hypothesis as an evolutionary mechanism driving song diversity in birds. An optimal balance between repetition and diversity may explain the singing style of many bird species and displays of other animals.

In many communication systems across a wide range of taxa, animals display ritualized behaviours that involve the repetition of stereotyped motor patterns, sometimes enhanced by elaborate ornaments[1]. While animal communication studies tend to focus on the exaggerated ornaments of many species, the study of ritualization in animal displays is starting to converge on the same theoretical framework across fields. The vigour and skill in the execution of motor displays are thought to be honest signals of whole-organism performance, reflecting the genetic quality and developmental history of an individual[2–4]. From fiddler crabs[5], to birds[6] or reptiles[7] and from human dancers[8,9] to athletes[10], consistency in the performance of repetitive motor tasks seems to convey relevant information and to be a target for natural and sexual selection.

Birdsong is a classic example of a ritualized display that modulates intra- and inter-sexual interactions[11]. The acoustic structure of song is shaped by natural selection, i.e. some songs transmit better than others through the environment[12,13], as well as by sexual

selection, as certain features indicate individual quality relevant in mate choice and conflict[14]. As a communication system, birdsong presents diversity, since individuals use different variants of song that share the same function, and ritualization, where the same pattern is repeated in a stereotyped fashion[14,15]. In birds, song diversity is thought to be a sexually selected trait[16], but this hypothesis has mixed support and it is unclear how it might provide an honest signal of quality[11,17]. There are multiple factors that could affect song diversity[15], but theoretically, a singing style that advertises song diversity should switch constantly between novel sound types and rarely repeat identical notes or songs. However, this is not usually the case and, as with other animal displays, birdsong is often a ritualized, repetitive display of stereotyped songs[18]. Although some authors have suggested that repetition and diversity respond to the selective pressures of intra- and inter-sexual selection, respectively[19], this hypothesis has not found wide support across species[14]. The paradox of apparent selection for both diversity and

[1]Lancaster Environment Centre, Lancaster University, Lancaster LA1 4YQ, UK. [2]Ecology and Environment Research Centre, Department of Natural Sciences, Manchester Metropolitan University, Manchester, UK. ✉e-mail: sierro.2.8@gmail.com; S.Dekort@mmu.ac.uk; i.hartley@lancaster.ac.uk

repetition is still puzzling[18] and suggests there are conflicting evolutionary forces at play.

One suggested mechanism explaining ritualization of animal displays is the role of motor performance in communication[2,6,18,20]. In songbirds, singing requires the execution of complex motor patterns[21], but most of these movements take place inside the body, hidden from view[21]. The song is the manifestation of these hidden movements in the shape of sound signals that can be assessed by competitors and potential mates. Higher motor performance of song is associated with higher social status[22], increased reproductive output[23,24], longevity[25,26], higher sexual attractiveness[27] and with fighting ability during territorial contests[25,28]. Each vocalisation in birdsong is produced by a stereotyped motor pattern performed by the phonatory organs[29] and repeating the same pattern with precision[20] is known as vocal consistency (Fig. 1). The repetition of identical notes within a song, typically found in trills, facilitates the assessment of vocal consistency, which reflects the motor performance skills of an individual[30,31].

Many bird species produce trilled songs in which the same note is repeated in quick succession[32]. Moreover, there is also repetition at a longer time scale as the entire song may be repeated along a singing bout. Repetition will increase the resilience of a communication system[33] but it might carry a cost in the form of receiver habituation, where the audience loses interest over time[34]. Habituation occurs when the response to a stimulus is reduced after several repetitions of the same stimulus[35]. The habituation hypothesis, also known as the monotony-threshold hypothesis[36], suggests that song diversity has evolved to reduce habituation in the audience. Presenting a series of different signals (i.e. song types), or longer silent pauses between subsequent song bouts, may reduce habituation during long singing displays. While multi-species analyses have failed to demonstrate a correlation between lower song diversity and longer silent pauses[37,38], there is empirical support for the habituation hypothesis as song type switching reduces habituation[39–41] and has a significant impact on the receiver during vocal interactions[42,43].

Here we show that vocal consistency, as a measure of precision in motor performance, is a fitness indicator in male blue tits and a key signal during mate choice, as females prefer songs of higher consistency. Moreover, we find strong support for the habituation hypothesis as a behavioural mechanism driving the evolution of diversity in birdsong. The evidence points towards a conflict between song repetition, to achieve the highest consistency, and song diversity to avoid habituation. We suggest that a balance between these two crucial aspects of song may resolve a long-standing paradox in studies of bird communication and explain the variability of singing styles between species.

## Results

### Male song variation, reproductive success, season and context

We found that male blue tits with higher vocal consistency (i.e., precise repetition of the same note type within a song) had higher reproductive success, as measured by clutch size (Fig. 2a, b, Tables 1, S1 and S2). Individual male song diversity (repertoire of song types) ranged from 2 to 7 (4.2 ± 1.3 song types per individual) and was not correlated with clutch size (Tables S1 and S2). Throughout the season, vocal consistency increased significantly within individuals from 13 weeks before egg laying (Fig. S1 and Table S3), peaked at dawn chorus during the receptive period of females and declined after that (Figs. 1 and 2c, d, Table S4). Based on the model predictions, male vocal consistency reached its seasonal maximum 4.9 days before the female partner began to lay eggs. This means that precision in male motor performance was highest at the start of the female receptive period (5 days before egg laying), when females made daily choices to mate, either with their social partner or with an extra-pair male. Vocal consistency also increased at another time scale, namely during consecutive repetition of the same song type during dawn song (Fig. 3, Table 2).

### Female choice experiment

All trials of the female choice playback experiment were performed during the egg-laying stage from the 22nd of April to the 4th of May 2020. We found no evidence to suggest that the experimental

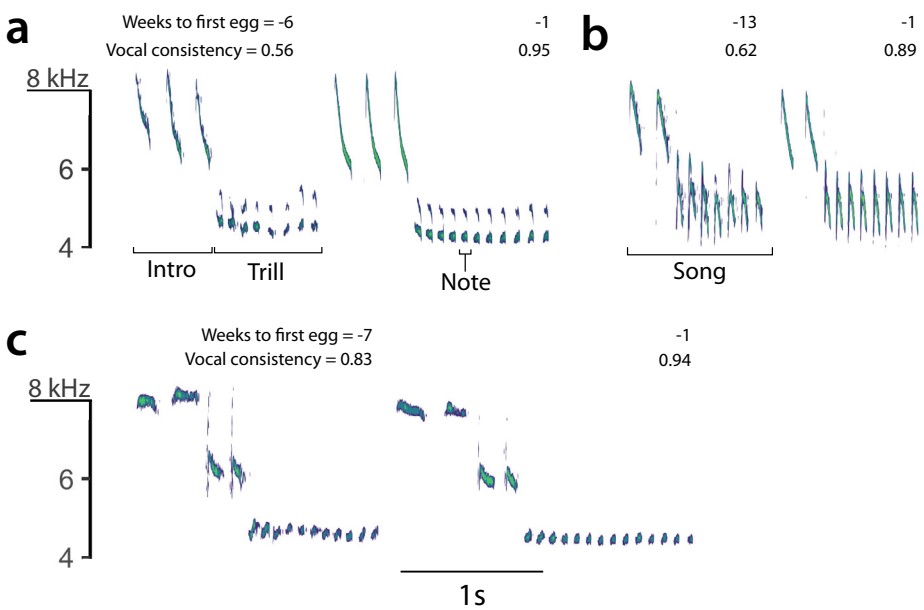

**Fig. 1 | Spectrograms of typical songs of blue tits.** Each of the three pairs of spectrograms (**a**–**c**) shows two songs recorded from the same individual male, one before the breeding period (left) and the second during the females' fertile period (right), with the frequency (kHz) in the Y-axis and time in seconds in the X-axis. Spectrograms (**a**, **b**) show annotations describing the basic structure of song in this species. The vocal consistency measured for each song is shown in the upper right corner, with 1 as the maximum consistency possible. The time when it was recorded during the season is also indicated as weeks in relation to the first egg date.

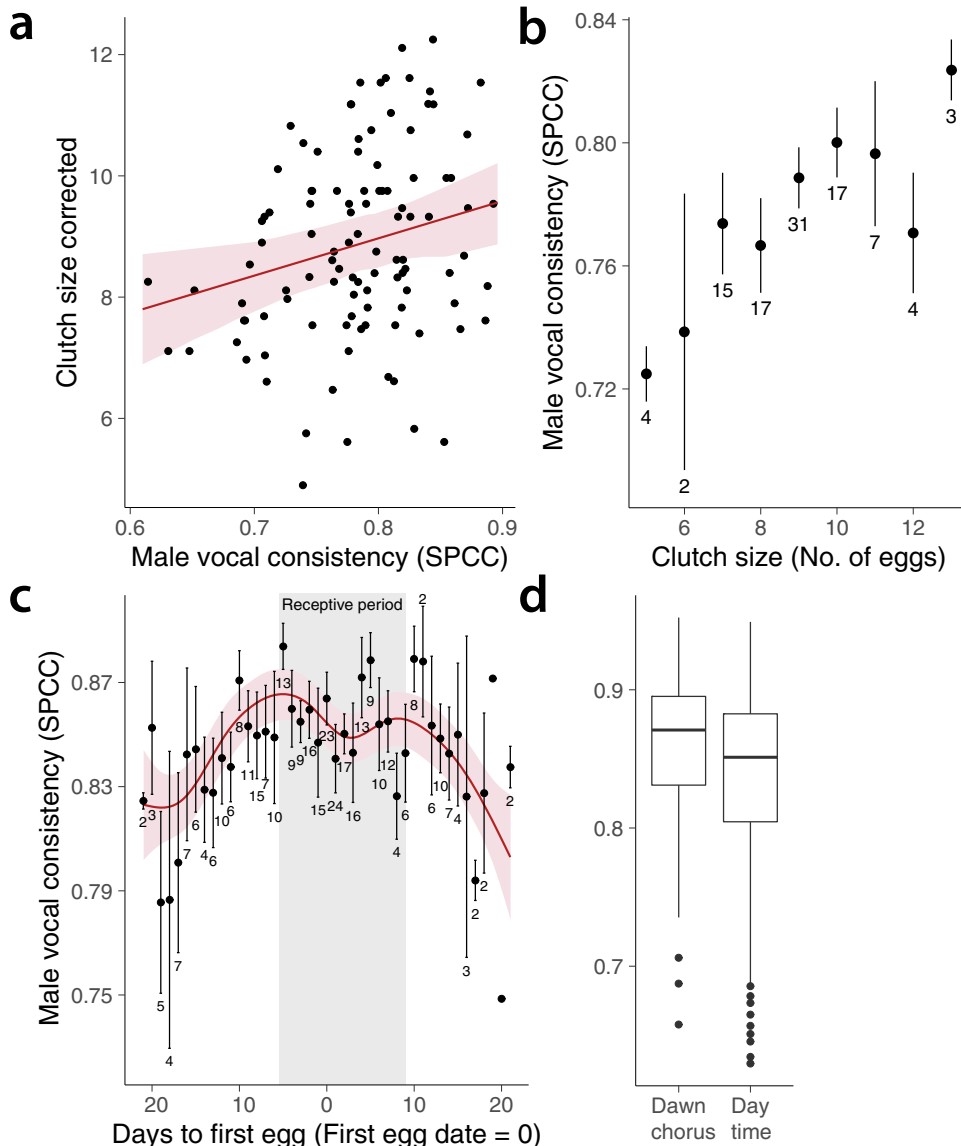

**Fig. 2 | Correlation of vocal consistency with reproductive success, season and context in blue tits.** Points in (**a**) show mean vocal consistency of each male per breeding season with the respective clutch size as number of eggs, corrected for the seasonal effect of date of first egg using the estimated coefficient from the model. Points in (**b**) show the male vocal consistency ± one standard error (SE) per clutch size category (number of males in each category shown underneath error bars). **c** The variation of male vocal consistency throughout the breeding period, with a temporal resolution of 1 day. Points indicate the mean ± SE of vocal consistency of all males recorded on each day, the sample size of number of males

recorded each day shown underneath error bars. In all plots, red lines trace the model predicted values, with the associated 95% confidence interval (CI) in pink shaded area. The grey area in the centre of (**c**) indicates the female receptive period. In (**d**), we see that male vocal consistency is significantly higher during dawn chorus (N = 90) than during day-time singing (N = 74), measured in a total of 95 birds recorded during the breeding period. Box and whisker plots showing median, upper and lower quartiles, and 1.5 interquartile range and outliers as points. Source data are provided in Supplementary Data 2.

**Table 1 | Estimates of the final model investigating impact of male song traits on reproductive success (clutch size)**

| Fixed effects | | | | | Random effects | |
|---|---|---|---|---|---|---|
| Variable | Estimate | T | 2.5% CI | 97.5% CI | Variable | Variance (SD) |
| Intercept | 9.597 | 35.698 | 9.077 | 10.12 | Individual | 0.61 (0.78) |
| Date of first egg (days from 1st of April) | −1.039 | −5.762 | −1.388 | −0.691 | Residual | 1.34 (1.16) |
| Vocal consistency | 0.367 | 2.639 | 0.098 | 0.635 | | |
| Year (2019) | −0.749 | −1.941 | −1.493 | −0.004 | | |
| Year (2020) | −1.725 | −4.037 | −2.552 | −0.897 | | |

For each fixed effect we show the model estimate, representing the size and direction of the effect, the T statistic and the 95% CI around the estimate. Vocal consistency had a positive significant effect on clutch size as the 95% CI do not overlap with zero. The marginal R squared ($R^2_m$) of the full model was 0.32 and the conditional R-squared represented ($R^2_c$) was 0.53. These indicate the goodness of fit of the model considering just the fixed effects ($R^2_m$) and including both fixed and random effects ($R^2_c$). The estimated variance and the standard deviation (SD) explained by the random effects are shown in the last two columns.

treatment influenced the breeding success of the females (Table S5). We recorded a total of 1295 individual calls, 94.8% of which were calls associated with copulation solicitation displays, 2.7% were churring, 0.5% were screaming calls while 2% could not be classified[44].

We found that female vocal response (proportion of playback bouts with at least one female vocal response) was significantly higher during the Song than during the Silence treatment (Song treatment = $0.25 \pm 0.17$, Silence treatment = $0.07 \pm 0.11$ proportion of bouts with female vocal response, $V = 3$, $P = 0.001$, 5% CI = $-0.28$, 95% CI = $-0.06$, Fig. 4a). Within the Song treatment, we found that the female vocal response was significantly positively correlated with vocal consistency of the playback stimulus but not with other song features such as song rate or trill length (Fig. 4b, Tables 3, S6 and S7),

Female vocal response showed behavioural signs of habituation to the continuous repetition of a song type, but there was clear dishabituation after a switch in song type and after a silent period, when vocal response increased significantly (Fig. 5, Table 4). Vocal response was significantly higher to the first songs presented after a silence gap than first songs after a song-type switch (Table 4).

## Discussion

We found that vocal consistency in blue tits showed a positive association with clutch size, a proxy for reproductive success. During continuous singing, males increased vocal consistency over the first 15

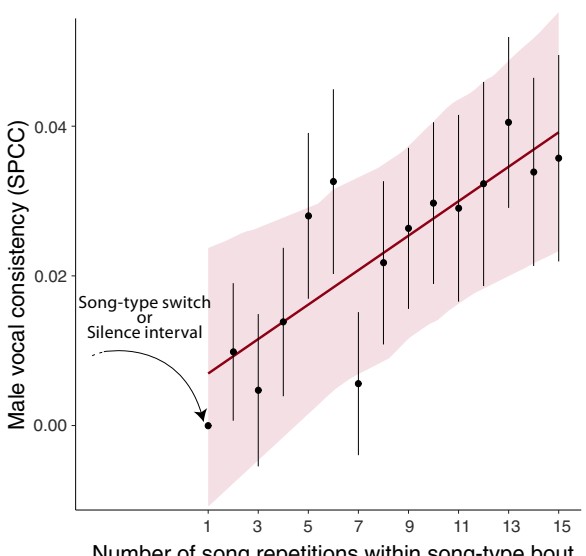

**Fig. 3 | Increase in vocal consistency within song in the first 15 repetitions of the whole song in a song-type bout.** Vocal consistency is standardized by subtracting the consistency measured in the first song per song-type bout (consistency of 1st repetition = 0). Points indicate mean values per song position across all song bouts for 18 individuals, with the associated SE. The red line indicates the predicted values of the model, with the associated 95% confidence interval in the shaded area. Source data are provided in Supplementary Data 2.

repetitions of the same song type. Vocal consistency also varied seasonally, with the peak in performance coinciding with the female receptive period, especially at dawn immediately before copulations[45,46]. Our playback experiment showed that females preferred consistent repetitions within songs but habituated quickly to the consecutive repetition of the same song type multiple times. Importantly, when playback song switched to a different song type, female response increased showing dishabituation due to song type switching. Our study indicates that male blue tit song is simultaneously selected for consistent repetition as well as for song diversity.

We propose that vocal consistency, as a measure of motor performance skill, is a fitness indicator in blue tits, based on multiple lines of evidence. First, we found a positive association between male vocal consistency and reproductive success. It is possible that this relationship is confounded to some extent by other variables (i.e. territory quality) and therefore it does not indicate female preference for higher vocal consistency in males. However, our results show that males that sing with higher vocal consistency will make a larger contribution to the next generation, indicating a positive selection pressure on vocal consistency in males. A potential limitation of this study is the fact that clutch size, as a measure of reproductive success, may include extra-pair young (EPY) not fathered by the social partner (the recorded male). One possibility, that would reduce the observed correlation, is that males with higher vocal consistency, which we found had larger clutches, also disproportionately lost paternity through extra-pair copulations (EPC). For this to be the case, there would need to be a higher proportion of EPY in larger clutches, however, our analysis on previous paternity data from this population shows that the opposite is true: larger clutches had a significantly lower proportion of EPY (see Methods). This is consistent with previous studies on fertilization in blue tits, which found that the likelihood of extra-pair fathered eggs declined with laying order[47–49], probably due to a decline in EPCs after the start of egg laying. As blue tits lay one egg per day, larger clutches require more days of egg laying, and therefore clutch size is not associated with more EPY in the brood[48,49]. Another possibility that would reduce the confidence in our measure of reproductive success would be if males with low vocal consistency (and small clutches) had a higher real reproductive success because they systematically gained more EPY in other nests. This possibility needs further exploration, although previous studies suggest that it seems unlikely that males paired with females that lay small clutches would systematically gain more paternity in other nests[23,50]. More work is necessary to reject these alternative explanations, but the structural analysis of EPY in relation to clutch size in our population, together with previous research on blue tits and other species, suggests that the observed pattern of larger clutch sizes in males of higher vocal consistency reflects an increased fitness for males with higher vocal consistency.

Second, the seasonal variation in male vocal consistency showed a marked congruence with female fertility. Male vocal consistency slowly increased during late winter, peaking right at the start of the female receptive period, that begins 5 days before egg laying, and declined after clutch completion. In temperate climates, the breeding season is relatively synchronized within species making it adaptive for

**Table 2 | Model estimates on the variation in vocal consistency as a function of song repetition within song-type bouts**

| Fixed effects | | | | | Random effects | |
|---|---|---|---|---|---|---|
| Variable | Estimate | T | 2.5% CI | 97.5% CI | Variable | Variance (SD) |
| Intercept | 0.005 | 0.552 | −0.012 | 0.021 | Individual | 0.00077 (0.028) |
| Number of repetitions | 0.002 | 3.863 | 0.001 | 0.003 | Residual | 0.0055 (0.074) |

For each fixed effect we show the model estimate, representing the size and direction of the effect, the T statistic and the 95% CI around the estimate. Vocal consistency increased significantly along with the number of repetitions in the song-type bout, as the 95% CI do not overlap with zero. The marginal R squared ($R^2_m$) of the full model was 0.02 and the conditional R-squared represented ($R^2_c$) was 0.14. The estimated variance and the standard deviation (SD) explained by the random effects are shown in the last two columns.

individuals to invest in exhibiting the most attractive signals during this time. The temporal synchrony between the female fertile period and the seasonal peak of male vocal consistency could be interpreted as circumstantial evidence for vocal consistency to be an important signal in mate choice or mate guarding[49,51–53]. This interpretation is also in line with our results from the playback experiment, as female sexual response was strongest to songs with high vocal consistency. From a proximate perspective, it may be that changes in brain structure associated with hormone cycles regulate the seasonal variation in male vocal consistency[54]. It is also possible that increased consistency is related with higher singing activity, as vocal practice leads to higher muscle performance in birds[55]. Other factors could play a role in shaping the structure of song throughout the year. For instance, increased ambient noise during spring (breeding season) in temperate

forests could lead to positive selection for songs of high vocal consistency, since the consistent repetition of a signal will improve signal transmission under noisy conditions[33].

Repetition of display signals is ubiquitous and allows for rapid assessment of song performance[56]. In our study, we found that males achieved their highest vocal consistency (precision) within song after multiple, consecutive repetitions of the same song type, i.e., a warming up effect[57]. Although females preferred higher vocal consistency within song, our experiment also showed that repetition of the entire song leads to habituation, decreasing sexual response. Hence, birdsong seems to be under evolutionary conflict and these opposing effects may result in a balance between repetition, to achieve the 'sexiest' song, and diversity (switching) between song types to reduce the effect of habituation. These results support the habituation hypothesis as a behavioural mechanism driving the evolution of song diversity[36,39]. Also, they can explain why most bird species repeat stereotyped song structures and do not sing in a way that advertises song diversity, for instance by continuously producing novel note or song types.

There is enormous variation in song diversity between songbird species, from the White's thrush *Zoothera aurea* with only one song type consisting of one or two notes (https://xeno-canto.org/species/Zoothera-aurea) to the common nightingale, *Luscinia megarhynchos* with a repertoire of hundreds of song types[58]. This interspecific variation in song displays is remarkable given that current theory assumes that song has largely the same functions across species. We believe that our study may provide a way out of that conundrum. We found evidence that both the silent gaps between repetitions and the switching between song types reduces habituation by females. It is plausible that multiple song types would further reduce habituation, compared to only two song types as we tested here. Displaying with higher diversity could also allow for a reduction of the silent pauses to avoid habituation, making communication more efficient. We hypothesise that the total duration of a singing display (i.e. a complete dawn song, an entire singing contest) will determine the importance of habituation in the evolution of signals, a factor that has not been considered in previous studies[37,38]. Hence, we propose that an optimal balance between consistent repetition and song diversity, considering other factors such as display duration and silence intervals, could partly explain the variation in singing styles across songbird species. This hypothesis predicts that longer song displays will be correlated with higher diversity across species, which can be tested in a comparative study. Other factors should be considered, as song diversity can be affected by the mating system[59], the habitat structure[60] and other evolutionary drivers[15].

While traditional views suggest that selective pressures associated with courtship and fighting target different signals, the study of motor performance in animal displays is gaining momentum as a unitary

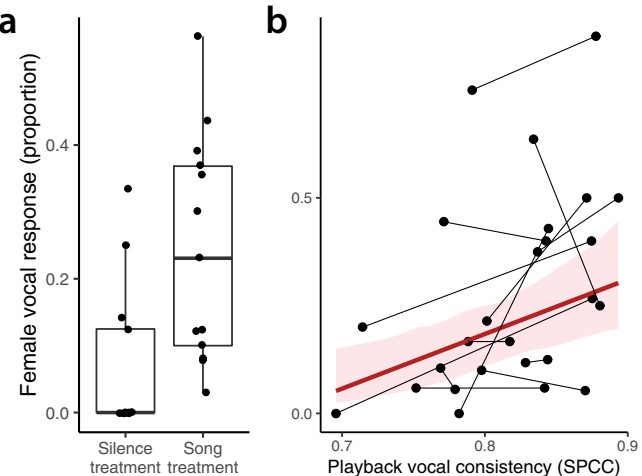

**Fig. 4 | Female vocal response during the playback experiment in the nest box.** Female response is shown as the proportion of song bouts where females produced at least one copulation solicitation call, in relation to the total number of bouts played while the female was inside the nest box. For the same 13 female subjects and for 442 bouts, **a** shows that female vocal response was significantly higher during the Song treatment than during the Silence treatment. Box and whisker plots showing median, upper and lower quartiles and 1.5 interquartile range. Points overlaid represent individual females, as the proportion of bouts with vocal response out of all bouts presented. **b** The significant, positive correlation between female vocal response and vocal consistency of playback song. In (**b**), points represent the proportion of bouts with vocal response out of all song bouts played for each song type during each trial. Black lines connect the response of the same female to both song types presented within the same trial. The red line represents the predicted values derived from the model, with the 95% CI interval in the pink shaded area. Source data are provided in Supplementary Data 2.

**Table 3 | Results from the Binomial GLMM exploring the variation in female vocal response as a function of the acoustic parameters of the playback song**

| Fixed effects | | | | | | | | | Random effects | |
|---|---|---|---|---|---|---|---|---|---|---|
| | Logit-transformed model estimates | | | Back-transformed model estimates | | | | | | |
| Variable | Estimate | 2.5% CI | 97.5% CI | Estimate | 2.5% CI | 97.5% CI | Z | Relative importance | Variable | Variance (SD) |
| Intercept | −1.406 | −1.925 | −0.887 | 0.197 | 0.127 | 0.292 | 5.312 | – | Individual | 0.46 (0.68) |
| Vocal consistency | 0.489 | 0.152 | 0.826 | 0.62 | 0.538 | 0.696 | 2.844 | 1 | | |
| Trill length (notes) | 0.06 | −0.216 | 0.64 | 0.515 | 0.446 | 0.655 | 0.4 | 0.28 | | |

For each factor, we present the model estimate, both the logit-transformed as well as the back-transformed, the 95% CI around the estimates, the Z statistic derived from Wald tests and the relative importance of that variable in the final model. The $R^2_m$ of the full model was 0.31 and the conditional $R^2_c$ was 0.76. The estimated variance and the standard deviation (SD) explained by the random effects is shown in the last two columns.

solution[4,20,21,61]. In birdsong, the precise and consistent execution of vocalisations is relevant in both fighting and mate attraction, ultimately providing an honest signal of fitness[56]. But in other taxa, similar concepts and terminology are being applied across a wide range of communication systems and disciplines: hermit crabs (*Pagurus bernhardus*) that fight with accuracy and precision are more successful[62,63]; fiddler crabs (*Uca tangeri*) show high individual consistency in their claw waving displays[5]; motor consistency has an important role in species recognition in Galápagos lava lizards[7]. In humans, physical training leads to higher consistency in muscle activation patterns[8,10] and high motor skills are both attractive[9] and more efficient during fights[64]. Moreover, subtle differences in a simple, repetitive motor

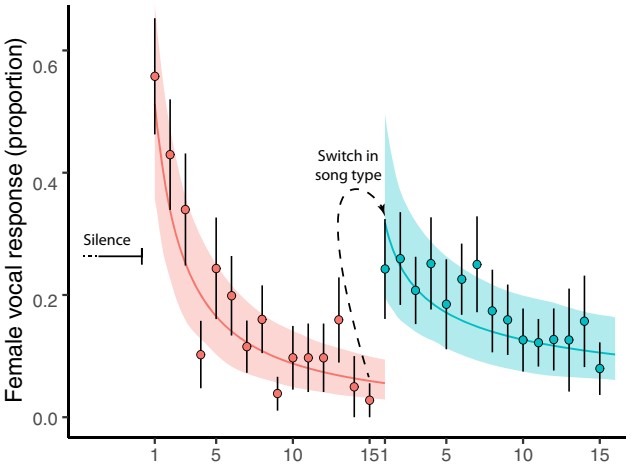

**Fig. 5 | Female vocal response per song position within playback song-type bouts.** The X-axis shows the sequential position of each song within a song-type bout. In this case, the first 15 repetitions are shown, matching the analysis of change in vocal consistency within song-type bout in male dawn song. The Y-axis shows the proportion of playback songs with at least one female vocalisation, out of all songs played per trial in that position. Each point represents the mean ± SE female vocal response per song position across all trials, summarizing the response of 13 females to 34 song-type bouts preceded by silence (in red) and 60 song-type bouts preceded by a song type switch (in blue). The lines show the predicted values for female vocal response derived from the model, with the associated 95% confidence intervals in shaded area. The decrease in response as the song is repeated indicates behavioural habituation of the females while the switch in song type elicited dishabituation. The data included song type switches between ten different song types across trials and in all possible directions, i.e., from song type A to song type B but also from song type B to song type A. Source data are provided in Supplementary Data 2.

task, such as finger tapping, is useful in the diagnosis of neurodegenerative diseases[65]. Overall, the evidence indicates that the precision or consistency of motor displays is a meaningful signal during communication across a wide range of contexts and taxa.

Our study sheds light on the apparent evolutionary paradox between repetition and diversity in birdsong. It supports the importance of motor performance skill in communication as a signal of fitness that is the target of mate choice, as well as the role of diversity of motor patterns to reduce habituation and enhance receiver response during long courtship displays. These results provide novel evidence to explain the evolution of multifaceted animal displays, especially in the role of motor performance in communication. Theoretically, the sexual selection pressures on repetition and diversity would lead to an optimal balance explaining the singing style of many bird species as well as other animal displays.

## Methods
All fieldwork involving the blue tits was approved by the Lancaster University animal welfare and ethical review board and licenced, where appropriate, by Natural England, and the British Trust for Ornithology. The study was conducted using a wild population of blue tits breeding in 110 boxes (mean ± SD: 67 ± 1.15 broods per year in three breeding seasons) near Lancaster University in the northwest of England (54.01′ N, 2.78′ W). This population is part of a long-term monitoring study[66]. Breeding birds were individually colour ringed after capture at winter feeding-stations or at the nest, caught using mist nets or traps in the nest boxes. Birds were measured (flattened, straightened wing length to nearest mm, tarsus length with foot bent down, to nearest 0.1 mm and head-bill length to nearest 0.1 mm) and weighed (to nearest 0.1 g)[67]. During the breeding season, individuals were sexed and aged in the hand based on plumage features and the presence of a brood patch or cloacal protuberance[68].

### Male song data
We studied the song of male blue tits (*Cyanistes caeruleus*), a well-established model species, to understand the role of repetition and diversity in birdsong. In the blue tit, as in many species, functional distinctions between songs and calls are not clear cut[69]. For this study, we used structural criteria to define blue tit song based on spectrograms reported in previous studies[70,71]. Song was defined as a vocalisation composed of a few introductory, high-pitched notes followed by a trill, always the last part of the song where a note is repeated several times in succession[71,72] (Fig. 1). Blue tits are discontinuous singers which repeat the same song structure with a fixed syntax, known as a song type, to make a song-type bout, before they switch to another song type[73]. Individuals learn several song types, and these are shared within a population. For this study, a note was defined as a

**Table 4 | Results from GLMM modelling female vocal response as a function of song position within a playback song-type bout and the interaction with song type switching**

| Fixed effects | | | | | | | | Random effects | |
|---|---|---|---|---|---|---|---|---|---|
| | Logit-transformed model estimates | | | Back-transformed model estimates | | | | | |
| Variable | Estimate | 2.5% CI | 97.5% CI | Estimate | 2.5% CI | 97.5% CI | Z | Variable | Variance (SD) |
| Intercept | 0.052 | −0.690 | 0.771 | 0.513 | 0.334 | 0.684 | 0.142 | Individual | 0.764 (0.87) |
| Song type switch | −0.813 | −1.46 | −0.167 | 0.307 | 0.188 | 0.458 | −2.47 | | |
| Song position | −1.039 | −1.32 | −0.764 | 0.261 | 0.210 | 0.318 | −7.32 | | |
| Song position: song type switch | 0.532 | 0.197 | 0.873 | 0.630 | 0.549 | 0.705 | 3.10 | | |

The table shows the model estimate, both the logit-transformed as well as the back-transformed, with the associated 95% CI around the estimate and the Z statistic. The intercept estimate shows the female vocal response at the start of a song-type bout preceded by silence. The significantly negative estimate of 'song position' indicates that the vocal response decreased significantly with increasing repetitions of the same song (habituation). The negative estimate of 'song type switch' indicates that the vocal response after a song type switch was significantly lower than at the start of a song-type bout preceded by silence. Finally, the positive, significant effect in the interaction between song position and song type switch indicates there was partial, but significant response recovery after a switch in song type.

continuous trace in the spectrogram (Fig. 1). From 2018 to 2020, we made audio recordings of individually identifiable males in the field using a Marantz PMD661 recorder (WAV format, 48 kHz sampling rate and 24-bit depth) and a Sennheiser ME67 microphone. Individuals were identified from coloured leg rings using binoculars (Nikon Prostaff 8x42). From January to May each year, we sampled songs of blue tits by making transects through the study site and conducting short waits at each nest box to identify and record any individual singing. The timing of recordings was adjusted to the singing activity of blue tits that shows strong seasonal variation[74,75]. During winter, recording times began after sunrise but the start of recording advanced as breeding period approached (mean first egg date of individual nests was the 22 April over the 3 years). Around 3 weeks before the start of egg laying, singing activity increases before sunrise, a phenomenon known as dawn chorus[24,76]. During dawn chorus, sampling transects were shorter and more time was spent recording each individual to record a complete dawn song. If ambient light was too low during the recording, we followed the same individual until light levels were high enough for identification using the rings.

We created a song database by manually selecting ten songs of the highest recording quality per song type, per date, per male. We focused on the trill section of the songs by selecting individual notes using Audacity[77], marking the start and end times of each note with the cursor. These markings were used to conduct acoustic analysis in R software (package 'tuneR'[78], package 'seewave'[79]; R Development Core Team, 2016[80]). As an important measure of motor performance, we measured vocal consistency[25] by estimating the mean acoustic similarity between subsequent renditions of the same note within trill using the Spectrogram Cross-Correlation Algorithm (SPCC)[81], which is an index of similarity between two sounds. We calculated the maximum correlation of every pair of notes with a maximum temporal offset of 20 ms and a temporal resolution of 1 ms. The spectrogram matrices were computed using an FFT algorithm with a window size of 512 samples, 90% overlap between successive windows and 'Hanning' window type. For any note, vocal consistency was the mean SPCC score of all pair-wise comparisons between that note and all other notes within the trill. For any song, vocal consistency was the mean vocal consistency of all notes within that trill. Also, we measured trill length, as the total number of notes per trill, and song rate, as the number of songs per minute. To measure song diversity, we classified song types visually within individuals based on song recordings made during two full dawn songs per individual. Based on previous research, this sampling strategy is sufficient to record the entire repertoire of each individual[45,82]. Overall, we built a library with 14 different song types in our population of 99 individual males. The singing style of blue tits, repeating the same song type multiple times to then switch to a different song type[73], allows for the somewhat objective visual categorization of song types within individuals based on the singing style and acoustic features.

## Automatic recordings at dawn

After mapping dawn song posts for each individual, we deployed autonomous recording units (ARUs), (Bioacoustic Audio Recorder -BAR-, Frontier Labs, WAV format, 48 kHz sampling rate and 16-bit depth) placed less than 1 m from the song post. The ARUs were in the same post for 3 to 5 days, but only 1 day with a long, continuous, high-quality recording of song was selected for analysis. This allowed us to record the dawn song of individual males to analyse the impact of repetition on vocal consistency. A song-type bout was defined as a string of continuous song where the same song structure (song type) is repeated many times (at least three) before stopping or switching to another song type. If there was a silent pause longer than 1 min, the next song was considered to belong to a new song-type bout, even if there was no switch in song type. Even though no human observer was there to read the leg colour-ring combination, we relied on song post

fidelity for identification, which was high even across years (personal observation). Furthermore, we used song cues of individual song-type repertoires to identify individual males, using manual recordings of known birds. Nevertheless, the fact that we recorded a specific bird was not as relevant as the fact that only one male was recorded per song post per dawn, as these recordings were not used to test for correlations between song characteristics and individual features or reproductive success. On the other hand, we selected song recordings with a continuous string of song near the microphone to ensure it was the same male singing. Movements of the singing bird with respect to the microphone were easy to identify in the sound recording. From these recordings, and given that the number of song repetitions is highly variable between bouts[73], we selected the first 15 repetitions of the same song in each song-type bout to measure vocal consistency within trill. Only song-type bouts with sufficient signal-to-noise ratio were selected and occasionally, some songs were left out if they overlapped extraneous sounds.

## Breeding data

During the breeding season, nest boxes were monitored every 3 days to determine occupancy, state of nest building, first egg date and clutch size. We defined the breeding period to start 3 weeks before the first egg of each nest, as birds start to perform behaviours related with reproduction, (i.e., increased territoriality around the nesting site, provide nesting material, etc.). In blue tits, extra-pair copulations (EPCs) are known to occur[51,83], but without the genetic data to measure the exact number of eggs fertilised by each male in the population, we chose clutch size (total number of eggs in the nest) as our proxy for reproductive success. Other measurements (e.g. fledging success) could be affected by chaotic events such as weather, desertions or predation, and would potentially increase the noise in the data. Since the measure of clutch size may include eggs fertilized by other males, we analysed previous paternity data from the same population[83] to understand whether our measure of reproductive success could be biased in support of our hypotheses. First, a two-sided, T-test was used to compare nests with and without extra-pair young (EPY) showing there was no statistical difference in clutch size (nests without EPY = $10.1 \pm 1.8$ mean ± standard deviation in clutch size, nests with at least one EPY = $9.8 \pm 2.3$ clutch size, 5% CI = −0.48, 95% CI = 1.03, $T = 0.72$, DF = 83.5, $P$ value = 0.47). This is consistent with results derived from a meta-analysis across 11 bird species[84]. Despite a small effect size (Cohen's $d = 0.13$), the statistical power was also relatively small (power = 0.1), so we need to be cautious in concluding that there was no difference in clutch size between nests with and without EPY[85]. However, as with a previous study in blue tits[48], we found that for nests with at least one EPY, clutch size was actually significantly negatively correlated with the proportion of EPY in the nest (5% CI = −0.61, 95% CI = −0.04, $T = -2.28$, $P = 0.028$, r = −0.36). This suggests that, for our hypotheses, clutch size is a conservative measure of reproductive success in male blue tits since those males with larger clutches would be likely to have an even higher overall fitness relative to males with small clutches. We could not test whether males with small clutches would gain fitness by fathering more EPY in other nests, however, this possibility seems unlikely based on previous studies in this species[23]. Thus, based on these results, we considered that clutch size was an appropriate measure of reproductive success (see 'Discussion').

During the laying period, female blue tits typically lay one egg per day and start incubation with the penultimate egg[86]. The female receptive period (when copulations take place) extends from 5 days before egg laying through to the completion of the clutch[51,87]. During this period, females roost inside the nest box and, at dawn, they interact vocally with their partners that are singing the dawn song outside near the box[44,52]. Dawn song is defined as a long, sustained display of song that begins around 30–90 min before sunrise and can last for more than 30 min, normally ending with copulation[45,46]. The

vocal interaction between male and female partners in this context indicates female stimulation, as the most common call produced by the female in the nest box is the copulation solicitation call (see 'Results'). We used this behaviour to test female preferences for song using playback, under the premise that females would show a stronger vocal response to male song to which they were more attracted[88,89]. The experiment was carried out during the egg laying period.

## Experimental design and playback stimuli

The experimental design consisted of the presentation of manipulated male song to females while they were inside the nest box during the dawn chorus. We attached a speaker (MIFA A1 Bluetooth speaker, 5 W, frequency response 80 Hz–18 kHz) to the outside bottom of the nest box facing towards the box (Figs. 6a and S2). Peak sound level inside the box was set to 60 dB(A) and the speaker was insulated to minimize sound leaking into the territory area, reducing the likelihood of the male interacting with the playback. We considered that the male reacted to the playback if it interrupted its own dawn song to approach the nest box while producing mobbing calls. We recorded audio inside the nest box using an AudioMoth[TM90] attached to the underside of the lid, which was an autonomous recording unit programmed to record from 1 h before to 1 h after sunrise (Fig. 6a, b). The equipment was installed on the nest box the day before the trial took place when the birds were not inside the nest, to avoid disturbance. On the morning of the trial, we placed a video camera on a tripod 10 m from the nest box to record the entrance hole. The trial began 30 min before sunrise and lasted 1 h or until the female exited the box (see Behavioural analysis). To ensure there was no alteration of the normal breeding behaviour, we compared the clutch size, brood size and fledging success of the females involved in the trial to the rest of the population after the breeding season. Previous trials suggested that females were likely to interact vocally with their own mate's song but not with synthetic or a different male's song (personal observation, see also ref. 91). There-

fore, to test female preferences, we selected two different song types of their own mate that varied in acoustic traits. We selected one song of each song type to study the variation in female response in relation to variation in vocal consistency between song types, controlling for a possible confounding effect of song rate and trill length[45] (i.e. Fig. S3). The male recordings used to build the playback stimuli were taken only some days before the trial was performed, during the breeding period. We chose a standard number of two song types per trial since it was the maximum number of song types recorded from all males. For our purposes, recording quality had to be extremely high, since the speaker was placed only centimetres away from the subject. In this set up, we suspected that even small recording artefacts could impact female response. Across trials, we used a total of ten song types which comprise the 91% of all identified song types (11) in the entire population across three years, after removing three song types with trills less than three notes long.

For each song type of each male, we selected one song in Audacity[77] and then used R software[78–80] to build a song-type bout by repeating it several times for a total time of 1:15 min. Song rate delivery was measured in ten songs of original recording for each song type per male, and this was the song rate used to build the song-type bout in each case. Two song-type bouts were placed in succession to make a string of playback song of 2:30 min. To this section, we added a silence interval of 2:30 min to make a playback round of 5 min. The entire playback stimulus lasted for 1 h and consisted of 12 playback rounds which means 12 silence bouts and 12 song bouts, each song bout with two song-type bouts, i.e., 24 song-type bouts in total. The order in which the song and the silence bout appeared within each round was randomized (Fig. 6b). In all cases, we introduced an interval of at least 5 min of silence at the beginning of the playback to leave some time for the female to relax after we approached the nest to start the playback. Please see Table S8 for operational definitions of playback structure.

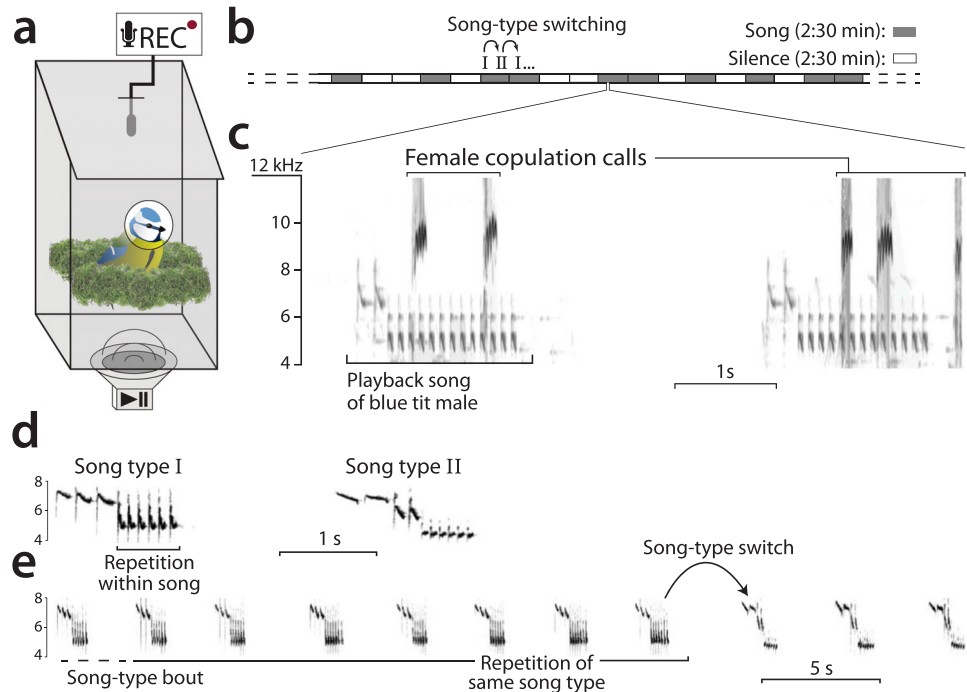

**Fig. 6 | Diagram of methods used in the female choice playback experiment.** In (**a**), a schematic overview of the equipment set up at the nest box, see also Fig. S5. In (**b**), a schematic timeline indicating the structure of a playback stimulus, alternating periods of song with silence. Spectrogram (**c**) shows an audio snapshot recorded inside the nest box during a playback trial, showing the vocal interaction of a subject female with the playback song, with frequency (kHz) in the Y-axis and time in seconds in the X-axis. Spectrograms (**d**) and (**e**) show the repetition at different time scales in a natural recording of a male singing. In (**d**) one note is repeated within song (this is where we measure vocal consistency), and in (**e**) the entire song is repeated within a song-type bout.

## Behavioural analysis of female choice experiment

We measured the vocal response of females in spectrograms (Figs. 6, S4 and S5) produced in Audacity software[77] (window type: 'Hanning', window length 1024 samples, 90% overlap and −80 dB range). Each vocalisation was classified using structural and functional criteria following Bijnens and Dhondt (1984) and Gorissen and Eens (2005) (see 'Results' and Figs. 6, S4 and S5). Copulation solicitation calls of blue tits are well defined in the literature[71,72] and have been recorded previously inside the nest box[44]. In blue tits, copulation calls show a characteristic acoustic structure beginning with an upward frequency sweep followed by quick frequency modulations between 8 and 12 kHz (Figs. 6, S4 and S5). Previously defined twittering and chattering calls[44] present very similar acoustic structure as copulation calls, varying gradually within a continuum (Figs. S4 and S5). In some cases, we could see how a call that looked like twittering gradually transformed into a genuine copulation call after few repetitions, and similarly in the case of chattering calls (Figs. S4 and S5). Hence, to avoid a subjective division of these call types, we used the three different call types to measure intensity of vocal response, as they were all associated with copulation solicitation displays (Figs. S4 and S5). Other types of calls such as churring or screaming were easily separated and these were not used to assess female arousal. Female vocal response was used to assess female preferences for song[88,89,92,93] and it was measured as a Binomial variable per song-type bout (female response: yes/no), marking yes if there was at least one vocalisation during a song-type bout of 1:15 min, considering solely calls associated with copulation solicitation displays. The complete presentation of the playback lasted for one hour or until the female exited the box, which was easily identified in the audio recordings[52] and confirmed with the video recordings. If the female entered the box again before the trial was over, we continued to take measurements of its vocal behaviour. Note that each trial consisted of the presentation of 24 song-type bouts, each of 1:15 min and more than 15 songs per bout (depending on the song rate of each song type). In most cases, the number of bouts presented to the female was lower since females could leave the box before the end of playback.

## Statistical analysis

All measures are presented as mean ± one standard deviation (SD), unless stated otherwise, and statistical analyses were carried out in R software 3.5.1[80]. Packages used in the statistical analysis included 'lme4'[94], 'MuMIn'[95] or 'gamm4'[96] and for data management and visualization we used 'stringr'[97], 'dplyr'[98] and 'ggplot2'[99]. Overall, we analysed a total of 7283 songs from 99 individual males during three years of data collection. From this data set, we selected a subset of males in each analysis, for which we had the required data to answer different questions.

## Statistical analysis of male song and reproductive success. To investigate the relationship between male song traits and reproductive success, we fitted a Linear Mixed effects Model (LMM) on the clutch size as a function of male vocal consistency (SPCC score), male song-type repertoire (number of song types recorded per individual). We also included the age of the female partner since it can affect reproductive success[100] and the Julian date of first egg in relation the 1st of April of the respective year, as it has been shown to influence clutch size[101,102]. The year of breeding (categorical) was included as a fixed effect, to control for inter-year variation in clutch size. Although the variation between years is not of interest to this study, it was included as a fixed effect since there were only three levels, three years, and therefore could not be included a as random effect[103]. Finally, we included the individual identity of males as a random effect to group together individual observations and avoid pseudo-replication. This model included a total of 106 observations, one for the clutch size of 106 nests across three years, with 5526 songs analysed from 77 individual males ($71.8 \pm 47.7$ songs per individual) for which we had

breeding data until the incubation stage and known age of the female partner.

Since the response variable was the clutch size per individual, which is one value per breeding season and repertoire size is one value per individual, we measured individual vocal consistency as the mean vocal consistency of all recordings per year ($54.1 \pm 32.3$ song analysed per individual per year in $3.9 \pm 1.8$ days of recording per individual per year). A one-way ANOVA conducted to test whether males were recorded in similar dates of the season showed there was no statistical difference in recording dates between individual males ($F_{(329, 76)} = 0.996$, $P$ value $= 0.49$). From the 77 individual males, 70 were recorded at least once during day-time singing and once during dawn chorus, while four individuals were recorded only during dawn chorus and three only during day-time. Dawn chorus was defined as all songs produced before 2 min after sunrise, as described by a previous study[24]. Hence our measure of vocal consistency per male per season was not context biased. Please note that repertoire size was measured always in at least two different dawn choruses, but some of these recordings served only to identify song types but not to measure vocal consistency.

**Statistical analysis of seasonal variation in male song.** To investigate large scale seasonal variation in vocal consistency within individuals, we selected 89 individual males that were recorded at least twice from 13 weeks before to 3 weeks after egg laying, in relation to individual breeding dates (week of first egg = 0). In this case, vocal consistency was normalized within individual, and values above zero mean that vocal consistency increased longitudinally within an individual. These 89 males, with 7006 songs ($78.7 \pm 54.1$ songs per individual), were recorded on $6.0 \pm 3.6$ (mean ± SD) different days, over a time span of $54.9 \pm 28.7$ days.

We used this data set to investigate the seasonal variation of male vocal consistency throughout the season by fitting a Generalized Additive Mixed Model (GAMM), a statistical approach suitable for the analysis of time series. Vocal consistency (normalized within individual), was the response variable as a function of weeks to first egg and singing context (dawn chorus vs. day-time singing), with week zero being the week when the first egg was laid for each specific male. Cross-validation was used to estimate the optimal amount of smoothing using cubic regression splines[104]. The singing context, a two-level categorical variable, was a fixed effect in the parametric side of the formula. This was to control for a possible context bias, since dawn chorus is a specific song display that occurs only during the breeding period[24,76]. As a random effect, we included the individual identity to avoid pseudo-replication and group individual observations together.

We then fitted a second GAMM model to investigate seasonal variation specifically during the breeding period at a finer time scale of days and not weeks, in relation to date of first egg per nest. In this second model, vocal consistency (not normalized) was the response variable, including all individuals recorded at least once during this period. This model used 5647 songs ($59.4 \pm 46.1$ songs per individual), recorded from 95 males. Each individual was recorded an average of $3.7 \pm 2.2$ different days within the breeding period with a mean of $14.7 \pm 10.8$ days between the first and the last recording. Of the 95 males, 69 were recorded both during day-time singing and during dawn chorus, 90 were recorded at least once during dawn chorus and 74 were recorded at least once during day-time singing.

**Statistical analysis of change in vocal consistency with song repetition.** We investigated the impact of continuous repetition of the same song type on vocal consistency within song by modelling the vocal consistency as a function of the position of song within the song-type bout (repetition number). To remove variation in vocal consistency due to season, time of day or between individuals, we standardized our measure of vocal consistency within each song-type bout

by subtracting the consistency measured in the first song to all other songs within the same song-type bout (i.e. consistency of first song = 0). We fitted an LMM model with the standardized vocal consistency as the response variable and the number of repetitions as a fixed effect, with the individual identity as a random effect. This model used 831 observations (individual songs), in 76 song-type bouts from 18 individuals (4.2 ± 2.4 bouts per individual). Note that we did not analyse exactly 15 songs per song-type bout since some bouts had fewer than 15 song repetitions and, in other cases, some songs were excluded as they overlapped extraneous sounds.

**Statistical analysis of female choice experiment.** For the female choice playback experiment, we conducted 15 playback trials. From these, one trial was discarded as we observed clear alteration in the male's behaviour following playback, which could disrupt female's reaction inside the box. From the remaining 14 trials, we removed one trial where the female did not produce any vocalisation during the entire trial. The remaining 13 females all produced at least one vocalisation associated with copulation solicitation displays (copulation, twittering or chattering) during the trial. Twelve of the 13 females left the box before the playback was over (14.2 ± 8.1 min before the end). Three females entered the box again before the playback was over and stayed for 6.1 ± 3.3 min until playback ended.

We compared the vocal response (proportion of bouts with at least one female call) between the Song treatment and the Silence treatment. The Song treatment includes all 2:30 min long bouts of playback song, and the Silence treatment includes all 2:30 min long sections of silence within the trial. We carried out a Wilcox signed-rank test with 26 paired observations of 13 individual females.

Within the Song treatment, we explored how female vocal response varied in relation to the acoustic features of playback song. To that end, we fitted a Binomial Generalized Linear Mixed-Effects Model (GLMM) where the number of successes vs. failures (bouts with/without female vocalization) was modelled as a function of vocal consistency, song rate and trill length of each song type within a playback trial. This model used 26 observations, one for each song type of 13 trials. The data summarized the responses of 13 females to 362 song-type bouts (27.1 ± 9.3 song-type bouts per trial), after removing those bouts that played while the female was absent from the box. Within and across trials, acoustic parameters varied between song types in different directions. For instance, the song type with the higher consistency could be the fastest trill in one case but the opposite in another trial. This allowed us to test the response of females to variation in consistency within trials, a paired design that increases the statistical power, while controlling for other confounding variables. In this model, we included female identity as a random effect to group individual observations together and avoid pseudo-replication.

We also tested the habituation effect in female vocal response by modelling the variation in female response as a function of the number of repetitions along each song-type bout. A song-type bout of 1:15 min had an average 17.3 ± 4.9 songs, depending on the song rate of each song type within trial. For example, a female that received seven playback song bouts could show a maximum of seven successes for each song position (i.e. number of repetition), meaning that the female vocalized in that specific song position in all song-type bouts presented. We also classified all song-type bouts into two categories, those that were immediately preceded by another song-type bout and those preceded by a silence bout. We fitted a Binomial GLMM with the number of successes vs. failures (with/without female response) as a function of the position of the song in the bout plus the full interaction with the type of song bout (preceded by silence or by song) (see Fig. 5). This allowed us to measure, and compare, the change in response (habituation) within a song-type bout but also explore the difference in response when the bout was preceded by silence or by a different song-type bout. Song position was log-transformed since the decline in vocal response was not linear along the bout, based on preliminary analysis. Finally, we included female identity as a random effect to group observations within subjects and avoid pseudo-replication. This model included 451 observations in 13 individual females, including a total of 94 song-type bouts with at least one female vocalization, 34 preceded by silence and 60 preceded by different song-type bout (song type switch).

In all (G)LMM models, we scaled and centred the predictor variables[105]. To validate all models, we confirmed that the residuals were homoscedastic and presented a normal distribution using diagnostic plots, in the case of linear models. We also tested for potential multicollinearity among the explanatory variables of the model by visual inspection of paired correlation plots and, in the case of (G)LMM also by estimating the Variance Inflation Factor (VIF) (*vif* function from 'car' package[106]) of the variables within the model. Multicollinearity among explanatory variables was assumed if VIF was greater than 3 on explanatory variables[104]. In the model to explore how male song correlated with reproductive success as well as the model exploring female vocal response in relation to playback song traits, we used an information theory approach computing all possible model combinations and ranking them by the Akaike Information Criterion corrected for small sample size (AICc)[107] to find which predictors were important in explaining the variation of the response variable. We selected all models that were within a ΔAICc <2 to compute the full average model as the final model[108,109]. We used the relative importance of each factor in the final model together with the coefficients and estimated confidence intervals (CI) with a threshold of 95%[109,110] to determine which variables had a significant effect in the response variable. Tables with all models below ΔAICc <7 are presented as supplementary material (Tables S2 and S7). In the model regarding female choice playback experiment, we did not carry a model selection procedure because the model structure was based on the experimental design.

**Reporting summary**
Further information on research design is available in the Nature Portfolio Reporting Summary linked to this article.

## Data availability
The authors declare that all data supporting the findings of this study are available within the supplementary information files of this publication. Raw data files are included as Supplementary Data 1. Data used to generate each figure are also provided as Supplementary Data 2.

## Code availability
The authors declare that the necessary programming code to conduct the statistical analysis presented in this study is available within the Supplementary Data 1.

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

## Acknowledgements
We want to thank Dr. Diego Gil, Dr. Stuart Sharp and Dr. Sarah Collins for their useful feedback on early versions of the manuscript. This research was supported by a research studentship funded by Lancaster University.

## Author contributions
S.K. provided the overall research concept; I.R.H., S.K. and J.S. conceived the study; J.S. collected behavioural data, sound recordings and designed the experimental study with inputs from S.K. and I.R.H.; Breeding and ringing data were collected by I.R.H. and J.S.; J.S. developed the code for the acoustic and the statistical analysis; J.S. wrote the paper with inputs from S.K. and I.R.H.

## Competing interests
The authors declare no competing interests.
