## [Peer Review File · Nature Communications]

Sexual selection for both diversity and repetition in birdsongEditorial Note: This manuscript has been previously reviewed at another journal that is not operating a transparent peer review scheme. This document only contains reviewer comments and rebuttal letters for versions considered at Nature Communications.

Reviewers' Comments:

Reviewer #1:

Remarks to the Author:

General Comments: This is a very well written, easy to read paper, which is much appreciated! It also encompasses a truly impressive amount of work and data collection. The study was very interesting, as were the results. Despite all of this, however, I am not sure that I completely agree with the author's conclusions.

- First, while I appreciate the limited space on a submission such as this, the introduction is written as if all birdsong reflects the singer's quality in some form or another. While I know that much of the bird literature does indeed focus on "honest signals" and indicator traits, there are many non-indicator hypotheses that have been proposed and supported with data from a wide range of systems. For example, there are many hypotheses for why distinct elements (i.e. increased diversity of elements) may be selected in communication displays and these need not be related at all to individual quality - e.g., increased 'information' or messages; multiple receivers; overcoming environmental heterogeneity; etc. As such, it isn't obvious to me why one might expect birdsong "to evolve to exhibit as much diversity as possibly during a display" (Line 20). Similarly, I can imagine non-indicator functions of increased consistency as well (e.g., noisy environment that requires the precise repetition of elements). I found that the immediate narrow focus on indicator hypotheses without consideration of other explanations (I will elaborate on some below) greatly diminished potential reach of this study, as it isn't clear how these findings fit within the broader hypotheses of complex signal evolution.

- More information on the mating system of blue tits would be great. In particular, in order to interpret the data, it is important to know if females engage in extra pair copulations. If they do, then paternity analyses would need to be done to confirm that the more consistent males are indeed the father of the counted offspring.

- A relationship between consistent repetition and egg number does not necessarily equate to consistent repetition being a fitness indicator (line 20). In addition to the possibility of extra pair copulations, females might invest more in males that sing more consistently. Or males may be altering their song in response to female feedback with females investing more in more responsive males.

- Could the decreased arousal by females not necessarily be habituation, but instead reflect the fact that there was no more dynamic interactions between her and the calling male? This likely reflects my lack of familiarity with the study system, but I would imagine that female-male interactions at the nest box would be quite dynamic. This would also explain why females re-engage (dishabituate in your interpretation) after a dynamic switch to a new song type.

- It seems like there are several reasons why vocal consistency might increase throughout the season. (i) Individuals have had an opportunity to practice. (ii) During the receptive period, females and males are potentially interacting more, providing opportunities for males to adjust signaling based on female feedback. (iii) Differences in the acoustic environment (e.g., more noise) requires adjusted signaling and selects for increased consistency....and more.

- I am not sure that I would call these "conflicting" selection forces, as there is always a variety of

factors that influence signal evolution (e.g., see the classic work by Guilford and Dawkins 1991 about strategic vs efficacy components). Even if the authors are correct in their interpretation of the function of repetition (a quality indicator reflecting motor performance) and diversity (dishabituation), these functions clearly work together. I guess I am not seeing how two song attributes that might be selected for different functions need to be in conflict.

- If diversity functions to reduce habituation, I would be interested to hear the authors thoughts on how much diversity one might predict/expect. Naively, I would imagine that only 2 song types would be necessary for dishabituation. How does this compare to what we see in nature?

Specific Comments:

- I was a little confused about sample sizes and whether or not the same birds were used across multiple samples. I see a subset of 76 males for inter-individual variation of vocal consistency, song diversity and clutch size; 89 males recorded in relation to individual breeding dates; and 95 males recorded from 3 weeks before to 3 weeks after first eggs. Are all of these different males?

- Similar to some of my previous comments, I was also curious about the potential role of female feedback in influencing male song performance, especially as it relates to a "warming up effect" (Line 173). This could also relate to the peak performance during the fertile period of the female. How much communication takes place between females and males and does this increase during the fertile period as well?

- Line 186 – it isn't clear what is meant by "flirting"

Reviewer #2:

Remarks to the Author:

This is a revised manuscript that I am reviewing for the first time. I first read and evaluated this draft of the manuscript on its own merits and only then read the previous reviews and the authors' responses to those reviews.

The key points made in this manuscript are that song consistency – that is, the similarity of repeated elements in a song – is a feature used by females in mate choice, and that there is a potential trade-off in singing between displaying consistency and displaying diversity (i.e., increased complexity within and/or between song types), leading to the question of how sexual selection may act on either trait. The blue tit is a good species in which to address these issues because males sing repertoires of song types (and thus could emphasize the display of diversity), and these song types are generally comprised of repeated acoustic elements (and could thus emphasize the display of consistency).

The field data presented here provide good correlative evidence that males sing with greater consistency when females are most likely to copulate. Earlier reviews raised very valid questions about the statistical nature of these analysis, which may be confounded by a number of factors, but the response to these reviews and the ensuing revisions in the present draft largely have addressed this set of concerns.

A remaining concern raised in the earlier review is that it is difficult to rule out the possibility that the effect of song consistency on female reproductive behavior (measured as clutch size) is simply a correlation driven by some other underlying factor, such as territory quality (or any other number of potential causal drivers). Data from the playback experiment provide some support for a causal relationship between song consistency and female response, although I do not find those data fully

convincing for reasons described below.

A related weakness is the lack of data on genetic parentage, as this species is known to engage in a high level of extra-pair fertilizations (EPFs). The authors suggest that variation in EPFs does not increase with clutch size, based on previously published data. However, given that clutch size on its own (to my knowledge) is not a typical measure of female choice (that is, that females when hearing a more preferred song from their social mate are provoked to lay more eggs than when hearing a less preferred song), this issue needs to be addressed more thoroughly to be convincing.

The playback experiment is very cleverly designed and does provide some evidence that vocal consistency of a female's social mate does change her behavior, at least as measured by her vocal responses. A concern raised in the previous review was to ask whether "copulation solicitation calls" reliably reflect female preference. I agree with the authors response on this point. These calls are often used as a measure of female preference and studies have shown that calling is associated with the propensity to actually copulate.

My concern with the playback experiment is that the comparison of response to high vs low consistency songs is based on playing back two different song types from the social mate's repertoire: one song type happening to have high vocal consistency and the other song type happening to have low vocal consistency. A much stronger design would be to manipulate one song type to have either a higher or lower consistency score, with the comparison being female response to the same song type heard under these two conditions. An even more ideal design would be to do this manipulation such that a song type is manipulated to have a higher consistency and also, separately, manipulated to have a lower consistency. This is not a fatal flaw, just a weakness with the experimental design that means one can't completely rule out some other, unknown factor associated with different song types affecting the outcome. This is especially a concern when the number of playback stimuli used is relatively small, as is the case here. All this said, it may be that I am incorrect in which stimuli were compared in the playback (e.g., line 126 refers to "manipulated" songs, but I couldn't find any other reference to how songs were manipulated). In any event, I found it difficult to find this information clearly stated in the methods, so it would be good to have this information presented more clearly).

Some other general comments:

I understand that the manuscript title has changed in response to comments in the previous review, but the current title is hard to understand. Is "trade-off" a better word to use than "conflict" here? Also, the word "promotes" does imply causality, which I understand was a concern in the earlier review.

I fully agree that the trade-off between displaying consistency versus displaying diversity is an interesting and important aspect of song and female choice to understand, but I think the manuscript over-simplifies the situation a bit by tacitly implying that these are the sole, or at least the dominant features of song that could influence female response. It is true that early work on sexual selection and song focused on song diversity, but there are other features of song that have been shown to influence female mating preferences, such as adherence to a local population dialect (e.g., Baker et al 1987 *Anim Behav* 35:1766), accuracy of song learning (reviewed in Lachlan 2012, *Am Nat* 180:751), or vocal performance (as a measure of the difficulty in producing a song, reviewed in Podos et al 2016 *Anim Behav* 116:203, which is different from the ability to produce a repeated feature of a song in a consistent fashion, as is analyzed in this manuscript). I realize that it can be hard to provide a broader background in a very short format journal, but for the non-specialist reader it would be important to briefly acknowledge the range of kinds of features female songbirds have been shown to respond to in the context of mate choice, even if this was just an opening clause such as "Among the several

features of male song that have been shown to influence female mate choice (refs), most attention has been paid to ..." Or something like that.

A few references didn't quite make sense to me:

Line 63: reference 20 on super-fast muscles is interesting, but does Rome directly support the point made here?

Line 65: reference 21 doesn't seem relevant here; in fact, this paper rules out the importance of motor performance. Here's a quote from that paper: "While the possibility exists that shifts in frequency are in response to motor constraints such as exhaustion from repeating songs at the same frequency (Lambrechts 1996), the fact that birds do not sing lower frequency dawn chorus songs any more often than the putatively more physiologically demanding high frequency songs suggests this is not the case."

Line 106: Reference 43 is a specific paper about ducks, and so doesn't provide very strong support for the point being made here, especially since ducks are quite different from songbirds in many aspects of their reproductive physiology and behavior.

Finally, I did not find the connection to human motor performance a useful addition to the manuscript because the connections here seem quite superficial, especially since references 8 and 10 are quite general and only reference 9 squarely supports the idea.

REVIEWER COMMENTS

Reviewer #1 (Remarks to the Author):

General Comments: This is a very well written, easy to read paper, which is much appreciated! It also encompasses a truly impressive amount of work and data collection. The study was very interesting, as were the results. Despite all of this, however, I am not sure that I completely agree with the author's conclusions.

- First, while I appreciate the limited space on a submission such as this, the introduction is written as if all birdsong reflects the singer's quality in some form or another. While I know that much of the bird literature does indeed focus on "honest signals" and indicator traits, there are many non-indicator hypotheses that have been proposed and supported with data from a wide range of systems. For example, there are many hypotheses for why distinct elements (i.e. increased diversity of elements) may be selected in communication displays and these need not be related at all to individual quality - e.g., increased 'information' or messages; multiple receivers; overcoming environmental heterogeneity; etc. As such, it isn't obvious to me why one might expect birdsong "to evolve to exhibit as much diversity as possibly during a display" (Line 20). Similarly, I can imagine non-indicator functions of increased consistency as well (e.g., noisy environment that requires the precise repetition of elements). I found that the immediate narrow focus on indicator hypotheses without consideration of other explanations (I will elaborate on some below) greatly diminished potential reach of this study, as it isn't clear how these findings fit within the broader hypotheses of complex signal evolution.

We would like to thank the reviewer for their positive feedback. In response to this comment, we have added further information regarding possible evolutionary pressures that may shape bird song, including factors other than quality indicators, please see L40, 47, 268 and L303

- More information on the mating system of blue tits would be great. In particular, in order to interpret the data, it is important to know if females engage in extra pair copulations. If they do, then paternity analyses would need to be done to confirm that the more consistent males are indeed the father of the counted offspring.

We have now moved part of the methods from the Supplementary Information to the main text (L93 & 149), in order to provide more information to the reader on the mating system of blue tits. Even though blue tits engage in extra-pair copulations, we have included an analysis which supports the idea that clutch size is an appropriate measure of reproductive success (L100). We now discuss this analysis and the limitations of our findings in a new paragraph (L225).

- A relationship between consistent repetition and egg number does not necessarily equate to consistent repetition being a fitness indicator (line 20). In addition to the possibility of extra pair copulations, females might invest more in males that sing more

consistently. Or males may be altering their song in response to female feedback with females investing more in more responsive males.

Our results indicate that males with higher vocal consistency will have a larger contribution to the next generation, which ultimately indicates selection for the traits present in those males. Therefore, those traits could be considered as fitness indicators. However, we agree in that the association between vocal consistency and clutch size may be affected by other factors and does not indicate female preference. Evidence for female preference for high vocal consistency is derived from other parts of the study. We have adjusted some sentences to be more careful in how we present our interpretation of the results derived from different parts of the study. Please see L225 and paragraph L255

- Could the decreased arousal by females not necessarily be habituation, but instead reflect the fact that there was no more dynamic interactions between her and the calling male? This likely reflects my lack of familiarity with the study system, but I would imagine that female-male interactions at the nest box would be quite dynamic. This would also explain why females re-engage (dishabituate in your interpretation) after a dynamic switch to a new song type.

As far as we know, there are very few studies that study the vocal interaction between the singing male and the female during dawn chorus in blue tits or other species. During three years of data collection, we observed that males engage in very long singing displays outside the box and it is only when the female exits the box that they interact physically, often copulating (L133 & 151). In this sense, our playback design seems to reflect the same level of interaction.

Habituation is defined as a “decrease in the strength of a naturally elicited behaviour that occurs through repeated presentations of the eliciting stimulus”¹. In our experiment we found that females responded naturally to the presentation of male blue tit song, but the strength of the response decreased after several repetitions of the same song type, which is consistent with the definition.

- It seems like there are several reasons why vocal consistency might increase throughout the season. (i) Individuals have had an opportunity to practice. (ii) During the receptive period, females and males are potentially interacting more, providing opportunities for males to adjust signaling based on female feedback. (iii) Differences in the acoustic environment (e.g., more noise) requires adjusted signaling and selects for increased consistency... and more.

We thank the reviewer for this comment and we have included some of these possible explanations when discussing seasonal variation in vocal consistency. Please see L265-272

- I am not sure that I would call these “conflicting” selection forces, as there is always a variety of factors that influence signal evolution (e.g., see the classic work by Guilford and Dawkins 1991 about strategic vs efficacy components). Even if the authors are correct in their interpretation of the function of repetition (a quality indicator reflecting motor performance) and diversity (dishabituation), these functions clearly work together. I guess I am not seeing how two song attributes that might be selected for different functions need to be in conflict.

In line with this comment and to avoid confusion with the classical example of conflicting evolutionary forces, we have modified the title and re-phrased some parts of the abstract and manuscript (L321 and 327). However, we suggest there is a conflict between repetition and diversity because males achieve the highest consistency after several repetitions (most attractive song), but repetition at this level also leads to lower female response (habituation). At the same time, we aim to reflect a history of paradoxical findings in the study of birdsong² which has often found a contradiction between high levels of repetition and large diversity in some species.

- If diversity functions to reduce habituation, I would be interested to hear the authors thoughts on how much diversity one might predict/expect. Naively, I would imagine that only 2 song types would be necessary for dishabituation. How does this compare to what we see in nature?

This is an interesting point we have now added a new paragraph in the discussion (L286). Here we suggest that, depending on the duration of the display, more diversity than two song types can be expected. Furthermore, we provide testable predictions for future research.

Specific Comments:

- I was a little confused about sample sizes and whether or not the same birds were used across multiple samples. I see a subset of 76 males for inter-individual variation of vocal consistency, song diversity and clutch size; 89 males recorded in relation to individual breeding dates; and 95 males recorded from 3 weeks before to 3 weeks after first eggs. Are all of these different males?

We have clarified the use of different sample size for different analyses in L92.

- Similar to some of my previous comments, I was also curious about the potential role of female feedback in influencing male song performance, especially as it relates to a “warming up effect” (Line 173). This could also relate to the peak performance during the fertile period of the female. How much communication takes place between females and males and does this increase during the fertile period as well?

Based on our experiment and previous studies³, females seem to be vocally active inside the nest, but little is known on how this interaction affects male (singing) behaviour. Outside the nest cavity, we have just recently shown that female blue tits sing frequently⁴, but little experimental research has been conducted on the function of song in females. Hence, we want to be cautious about discussing a possibility for which we have little evidence.

- Line 186 – it isn’t clear what is meant by “flirting”
The term flirting has been changed to “mate attraction”

Reviewer #2 (Remarks to the Author):

This is a revised manuscript that I am reviewing for the first time. I first read and evaluated this draft of the manuscript on its own merits and only then read the previous reviews and the authors' responses to those reviews.

The key points made in this manuscript are that song consistency – that is, the similarity of repeated elements in a song – is a feature used by females in mate choice, and that there is a potential trade-off in singing between displaying consistency and displaying diversity (i.e., increased complexity within and/or between song types), leading to the question of how sexual selection may act on either trait. The blue tit is a good species in which to address these issues because males sing repertoires of song types (and thus could emphasize the display of diversity), and these song types are generally comprised of repeated acoustic elements (and could thus emphasize the display of consistency).

The field data presented here provide good correlative evidence that males sing with greater consistency when females are most likely to copulate. Earlier reviews raised very valid questions about the statistical nature of these analysis, which may be confounded by a number of factors, but the response to these reviews and the ensuing revisions in the present draft largely have addressed this set of concerns.

A remaining concern raised in the earlier review is that it is difficult to rule out the possibility that the effect of song consistency on female reproductive behavior (measured as clutch size) is simply a correlation driven by some other underlying factor, such as territory quality (or any other number of potential causal drivers). Data from the playback experiment provide some support for a causal relationship between song consistency and female response, although I do not find those data fully convincing for reasons described below.

A related weakness is the lack of data on genetic parentage, as this species is known to engage in a high level of extra-pair fertilizations (EPFs). The authors suggest that variation in EPFs does not increase with clutch size, based on previously published data. However, given that clutch size on its own (to my knowledge) is not a typical measure of female choice (that is, that females when hearing a more preferred song from their social mate are provoked to lay more eggs than when hearing a less preferred song), this issue needs to be addressed more thoroughly to be convincing.

We agree in that a clutch size is not a good indicator of female choice and we have adjusted the text to make this clear. Now, the first paragraph of the discussion strictly describes a summary of the main results. Then, we have added two new paragraphs where we derive specific evidence from each part of the study. In these paragraphs, we try to make clear that the association between male vocal consistency and clutch size may be mediated by other factors and does not reflect female choice but that it indicates positive selection (L227). Then, in the next paragraph, we interpret that the playback response indicates female preference for higher vocal consistency in male song (L255).

The playback experiment is very cleverly designed and does provide some evidence that vocal consistency of a female's social mate does change her behavior, at least as

measured by her vocal responses. A concern raised in the previous review was to ask whether “copulation solicitation calls” reliably reflect female preference. I agree with the authors response on this point. These calls are often used as a measure of female preference and studies have shown that calling is associated with the propensity to actually copulate.

My concern with the playback experiment is that the comparison of response to high vs low consistency songs is based on playing back two different song types from the social mate’s repertoire: one song type happening to have high vocal consistency and the other song type happening to have low vocal consistency. A much stronger design would be to manipulate one song type to have either a higher or lower consistency score, with the comparison being female response to the same song type heard under these two conditions. An even more ideal design would be to do this manipulation such that a song type is manipulated to have a higher consistency and also, separately, manipulated to have a lower consistency. This is not a fatal flaw, just a weakness with the experimental design that means one can’t completely rule out some other, unknown factor associated with different song types affecting the outcome. This is especially a concern when the number of playback stimuli used is relatively small, as is the case here. All this said, it may be that I am incorrect in which stimuli were compared in the playback (e.g., line 126 refers to “manipulated” songs, but I couldn’t find any other reference to how songs were manipulated). In any event, I found it difficult to find this information clearly stated in the methods, so it would be good to have this information presented more clearly).

We agree with the referee in this argument, but we could not implement the ideal experimental design for practical reasons.

During pilot trials for the playback experiment, we observed that the presentation of song did not elicit a response from females unless this song was her mate’s song. In the same way, strong manipulation of song structure could modify the identity cues of the song that would make it recognizable as her mate’s song. For these reasons, we decided to use the natural variation in vocal consistency between song types of the social male to carry out our experiment. Please see L689 in Supplementary Information. We have also added a sentence in the main text to provide more detail of the playback process in the main text (159, 174).

To deal with variation between different song types, we made sure that we used nearly all song types recorded in our population across different trials and we controlled for possible confounding factors by including other variables that differed between song types as predictors in our statistical model (L881).

Some other general comments:

I understand that the manuscript title has changed in response to comments in the previous review, but the current title is hard to understand. Is “trade-off” a better word to use than “conflict” here? Also, the word “promotes” does imply causality, which I understand was a concern in the earlier review.

Thank you for this comment. We have now returned to our previous title with a modification.

I fully agree that the trade-off between displaying consistency versus displaying diversity

is an interesting and important aspect of song and female choice to understand, but I think the manuscript over-simplifies the situation a bit by tacitly implying that these are the sole, or at least the dominant features of song that could influence female response. It is true that early work on sexual selection and song focused on song diversity, but there are other features of song that have been shown to influence female mating preferences, such as adherence to a local population dialect (e.g., Baker et al 1987 Anim Behav 35:1766), accuracy of song learning (reviewed in Lachlan 2012, Am Nat 180:751), or vocal performance (as a measure of the difficulty in producing a song, reviewed in Podos et al 2016 Anim Behav 116:203, which is different from the ability to produce a repeated feature of a song in a consistent fashion, as is analyzed in this manuscript). I realize that it can be hard to provide a broader background in a very short format journal, but for the non-specialist reader it would be important to briefly acknowledge the range of kinds of features female songbirds have been shown to respond to in the context of mate choice, even if this was just an opening clause such as “Among the several features of male song that have been shown to influence female mate choice (refs), most attention has been paid to ...” Or something like that.

Thank you for this comment, we have added two new sentences in the paragraph 2 of the introduction to present a broader view as suggested (see L41, 48 and 72).

A few references didn't quite make sense to me:

Line 63: reference 20 on super-fast muscles is interesting, but does Rome directly support the point made here?

We have removed this reference from the manuscript as it was indeed misplaced. We have replaced it with: “Goller, F. (2021). Vocal athletics—from birdsong production mechanisms to sexy songs. Animal Behaviour”

Line 65: reference 21 doesn't seem relevant here; in fact, this paper rules out the importance of motor performance. Here's a quote from that paper: “While the possibility exists that shifts in frequency are in response to motor constraints such as exhaustion from repeating songs at the same frequency (Lambrechts 1996), the fact that birds do not sing lower frequency dawn chorus songs any more often than the putatively more physiologically demanding high frequency songs suggests this is not the case.”

We have removed this reference from the manuscript

Line 106: Reference 43 is a specific paper about ducks, and so doesn't provide very strong support for the point being made here, especially since ducks are quite different from songbirds in many aspects of their reproductive physiology and behavior.

We have removed this reference and the statement from the manuscript

Finally, I did not find the connection to human motor performance a useful addition to the manuscript because the connections here seem quite superficial, especially since references 8 and 10 are quite general and only reference 9 squarely supports the idea.

We agree with the reviewer that some of the references to human behaviour may come from distantly related fields of research. Nevertheless, we believe that the conceptual approach and the methodological process are more closely related than it may appear. Please see a newly added reference to a study of motor skill during fights in humans (boxing), L316. Similar measures of motor skill as used here can be applied to assess

many different types of animal displays and some of the findings made in studies of human dancing or boxing lead to similar conclusions as ours. For these reasons we have opted for keeping these references in our manuscript.

1. Bouton M. Learning and behavior: a contemporary synthesis (Sinauer, Sunderland, MA). (2007).
2. Price JJ. Why is birdsong so repetitive? Signal detection and the evolution of avian singing modes. *Behaviour* **150**, 995-1013 (2013).
3. Gorissen L, Eens M. Complex female vocal behaviour of great and blue tits inside the nesting cavity. *Behaviour* **142**, 489-506 (2005).
4. Siervo J, de Kort SR, Riebel K, Hartley IR. Female blue tits sing frequently: a sex comparison of occurrence, context, and structure of song. *Behav Ecol*, arXiv:2204.04404 (2022).

Reviewers' Comments:

Reviewer #1:

Remarks to the Author:

I applaud the authors on their thorough and thoughtful response to the previous round of reviews. All of the major concerns that I had have been addressed and I think the currently revised manuscript will be of interest to a wide audience.

Reviewer #2:

Remarks to the Author:

The current revision of this manuscript addresses many of the issues raised in the earlier review, clarifying a number of important points. Two points remain less well resolved, however.

The revised manuscript now provides more support for using clutch size as a proxy for reproductive success, presenting an analysis which suggests that this proxy is unlikely to be biased by EPFs (a major issue raised in the earlier review). The revised manuscript also does a good job of articulating the caveats associated with the use of this proxy. The fact that there are a number of caveats to consider, however – that is, other factors that could be responsible for the relationship between vocal consistency and clutch size – does weaken the overall argument. This is not a fatal flaw, just a weakness in the case being made here.

The playback experimental design which compares response to two different song types (one of low consistency vs one of high consistency) is not ideal, as pointed out in the earlier review. The revised manuscript now points out the constraints that require this design, and it also states that using nearly all song types recorded in the population makes it possible to statistically control for inherent song type differences, which is helpful. However, the explanation provided in the supplementary methods (line 880 and following) is unclear, at least to me. Specifically, the sentence “Within and across trials, acoustic parameters differed between song types along a continuous gradient, allowing us to test the response to this variation across and within females, increasing the statistical power.” should be unpacked and clarified.

This revision includes a new paragraph (line 285 and following) in response to another reviewer’s query about habituation. This paragraph confused me, however, in part because it isn’t clear what is meant by “total duration of a song display.” Does this refer to the length of a single song or song type, or to the length of a bout of singing? Some species, like great reed warblers, have repertoires of syllable type and present these in long continuous strings, whereas other species, like nightingales, sing large repertoires of song types with each song type being multi-parted and different song types clearly separated by silent periods. How does the idea presented in this paragraph map onto this kind of difference? Overall, without this idea being spelled out more completely and more clearly, it is not clear this adds to the main argument of this paper.

Reviewer #1

I applaud the authors on their thorough and thoughtful response to the previous round of reviews. All of the major concerns that I had have been addressed and I think the currently revised manuscript will be of interest to a wide audience.

Reviewer #2

The current revision of this manuscript addresses many of the issues raised in the earlier review, clarifying a number of important points. Two points remain less well resolved, however.

The revised manuscript now provides more support for using clutch size as a proxy for reproductive success, presenting an analysis which suggests that this proxy is unlikely to be biased by EPFs (a major issue raised in the earlier review). The revised manuscript also does a good job of articulating the caveats associated with the use of this proxy. The fact that there are a number of caveats to consider, however – that is, other factors that could be responsible for the relationship between vocal consistency and clutch size – does weaken the overall argument. This is not a fatal flaw, just a weakness in the case being made here.

We agree in that the correlation between male song consistency and clutch size may be related or associated with other variables, either other male traits or environmental factors associated with each male's territory. However, we believe that this does not weaken the argument being made in considering male vocal consistency as a fitness indicator since it is associated directly or indirectly with reproductive success and should therefore be under positive selection.

The playback experimental design which compares response to two different song types (one of low consistency vs one of high consistency) is not ideal, as pointed out in the earlier review. The revised manuscript now points out the constraints that require this design, and it also states that using nearly all song types recorded in the population makes it possible to statistically control for inherent song type differences, which is helpful. However, the explanation provided in the supplementary methods (line 880 and following) is unclear, at least to me. Specifically, the sentence “Within and across trials, acoustic parameters differed between song types along a continuous gradient, allowing us to test the response to this variation across and within females, increasing the statistical power.” should be unpacked and clarified.

This has been now clarified. Please see L572 of SI file.

This revision includes a new paragraph (line 285 and following) in response to another reviewer's query about habituation. This paragraph confused me, however, in part because it isn't clear what is meant by “total duration of a song display.” Does this refer to the length of a single song or song type, or to the length of a bout of singing? Some species, like great reed warblers, have repertoires of syllable type and present these in long continuous strings, whereas other species, like nightingales, sing large repertoires of song types with each song type being multi-parted and different song types clearly separated by silent periods. How does the idea presented in this paragraph map onto this kind of difference? Overall, without this idea being spelled out more completely and more clearly, it is not clear this adds to the main argument of this paper.

It may help to clarify this point if we change “song display” by “singing display”, as we refer to the total duration that a bird is singing within a particular behavioural context or interaction. We aim to include all kind of situations regardless of specific variations in species-specific singing behavior. We have clarified this in L230